# A Spectral Nonlocal Block for Neural Networks

## Abstract

The nonlocal network is designed for capturing long-range spatial-temporal dependencies in several computer vision tasks. Although having shown excellent performances, it needs an elaborate preparation for both the number and position of the building blocks. In this paper, we propose a new formulation of the nonlocal block and interpret it from the general graph signal processing perspective, where we view it as a fully-connected graph filter approximated by Chebyshev polynomials. The proposed nonlocal block is more efficient and robust, which is a generalized form of existing nonlocal blocks (e.g. nonlocal block, nonlocal stage). Moreover, we give the stable hypothesis and show that the steady-state of the deeper nonlocal structure should meet with it. Based on the stable hypothesis, a full-order approximation of the nonlocal block is derived for consecutive connections. Experimental results illustrate the clear-cut improvement and practical applicability of the generalized nonlocal block on both image and video classification tasks.

## 1 Introduction

Capturing the long-range spatial-temporal dependencies is crucial for the Deep Convolutional Neural Networks (CNNs) to extract discriminate features in vision tasks such as image and video classification. However, the traditional convolution operator only focuses on processing local neighborhood at a time. This makes the CNNs need to go deeper with convolutional operations to enlarge the receptive fields, which lead to higher computation and memory. Moreover, going deeper cannot always increase the effective receptive fields due to the Gaussian distribution of the kernel weight (Luo et al. (2016)). To eliminate this limitation, some recent works focus on designing the network architecture with wider and well-designed modules to catch the long-range dependencies such as (Peng et al. (2017), Chen et al. (2017), Zhao et al. (2017)). Although having larger receptive fields, these modules still need to be applied recursively to catch the dependencies of the pairs in large distances.

Inspired by the classical non-local means method in image denoising, Wang et al. (2018) proposes the nonlocal neural network which uses the nonlocal (NL) block to concern the "full-range" dependencies in only one module by exploring the correlations between each position and all other positions. In the NL block, the affinity matrix is first computed to represent the correlations between each position pair. Then the weight means of features are calculated based on the affinity matrix to refine the feature representation. Finally, the residual connection is added to the refined feature map. Due to its simplicity and effectiveness, the nonlocal block has been widely used in image and video classification (Wang et al. (2018); Yue et al. (2018); Tao et al. (2018); Chen et al. (2018)), image segmentation (Huang et al. (2018); Yue et al. (2018); Wang et al. (2018)) and person re-identification (Liao et al. (2018); Zhang et al. (2019)) recently.

However, due to the complexity of the affinity matrix, the nonlocal block [1] needs much more computational effort and is sensitive to its number and position in the neural network (Tao et al. (2018)). Some works solve the first problem by simplifying the calculation of the affinity matrix such as Huang et al. (2018), He et al. (2019), Yue et al. (2018), Chen et al. (2018). Only a few works try to solve the second problem which limits the robustness of the nonlocal network [2]. Tao et al. (2018)

---

[1] The nonlocal block is composed of a nonlocal operator and a residual connection
[2] The nonlocal network is composed of several nonlocal blocks

proposes the nonlocal stage (NS) block which concerns the diffusion nature and maintains the same affinity matrix for all the nonlocal units in the NS block. Comparing with the NL block, the NS block is insensitive to the numbers and allows deeper nonlocal structure. However, the deeper nonlocal structure of NS block increases the complexity and do not have a remarkable improvement.

In this work, we focus on elaborating a robust nonlocal block which is more flexible when using in the neural network. We prove that the nonlocal operator in the nonlocal block is equivalent to the Chebyshev-approximated fully-connected graph filter with irrational constraints that limits its liberty for learning. To remove these irrational constraints, we propose the Spectral-based Nonlocal (SNL) block which is more robust and can degrade into the NL and NS with specific assumptions. We also prove that the deeper nonlocal structure satisfies the stable hypothesis with the help of steady-state analysis. Based on this hypothesis, we give the full-order approximated spectral nonlocal (gSNL) block which is well-performed for deeper nonlocal structure. Finally, we add our proposed nonlocal blocks into the deep network and evaluate them on the image and video classification tasks. Experiments show that the networks with our proposed blocks are more robust and have a higher accuracy than using other types of nonlocal blocks. To summarize, our contributions are threefold:

- We propose a spectral nonlocal (SNL) block as an efficient, simple, and generic component for capturing long-range spatial-temporal dependencies with deep neural networks, which is a generalization of the classical nonlocal blocks.

- We propose the stable hypothesis, which can enable the deeper nonlocal structure without an elaborate preparation for both the number and position of the building blocks. We further extend SNL into generalized SNL (gSNL), which can enable multiple nonlocal blocks to be plugged into the existing computer vision architectures with stable learning dynamics.

- Both SNL and gSNL have outperformed other nonlocal blocks across both image and video classification tasks with a clear-cut improvement.

## 2 PRELIMINARY

**Nonlocal block** The NL block consist of NL operator with residual connection and is expressed as:

$$\mathbf{Y} = \mathbf{X} + \mathcal{F}(\mathbf{A}, \mathbf{Z}) \quad \text{with} \quad \mathbf{Z} = \mathbf{X}\mathbf{W}_g, \tag{1}$$

where $\mathbf{X} \in \mathbb{R}^{N \times C_1}$ is the input feature map, $\mathcal{F}(\mathbf{A}, \mathbf{Z})$ is the NL operator, $\mathbf{Z} \in \mathbb{R}^{N \times C_s}$ is the transferred feature map that compresses the channels of $\mathbf{X} \in \mathbb{R}^{N \times C_1}$ by a linear transformation with kernel $\mathbf{W}_g \in \mathbb{R}^{C_1 \times C_s}$. Here $N$ is the number of positions. The affinity matrix $\mathbf{A} \in \mathbb{R}^{N \times N}$ is composed by pairwise correlations between pixels.

In the NL block, the NL operator explores the "full-range" dependencies by concerning the relationships between all the position pairs:

$$\mathcal{F}(\mathbf{A}, \mathbf{Z}) = \mathbf{A}\mathbf{Z}\mathbf{W} \quad \text{with} \quad \mathbf{A} = (a_{ij})_{N \times N}, \quad A_{ij} = f(\mathbf{X}_{i,:}, \mathbf{X}_{j,:}), \tag{2}$$

where $\mathbf{W} \in \mathbb{R}^{C_s \times C_1}$ is the weight matrix of a linear transformation. $f(\cdot)$ is the affinity kernel which can adopt the "Dot Product", "Traditional Gasuassian", "Embedded Gasussian" or other kernel matrix with a finite Frobenius norm.

**Nonlocal stage** To make the NL operator follow the diffusion nature that allows deeper nonlocal structure (Tao et al. (2018)), the nonlocal stage (NS) operator uses the graph laplacian $\mathbf{L} = \mathbf{D}_A - \mathbf{A}$ to replace the affinity matrix $\mathbf{A}$ in the NL operator:

$$\bar{\mathcal{F}}(\mathbf{A}, \mathbf{Z}) = (\mathbf{A} - \mathbf{D}_A)\mathbf{Z}\mathbf{W} \quad \text{with} \quad \mathbf{D}_A = diag(d_i), \tag{3}$$

where $\bar{\mathcal{F}}(\mathbf{A}, \mathbf{Z})$ is the NS operator. $d_i = \sum_j a_{ij}$ is the degree of node $i$. Moreover, when adding multiple blocks with the same affinity matrix $\mathbf{A}$ and replacing the NL operator by the NS operator, these consecutively-connected blocks become the NS block. We called these nonlocal blocks in the NS block as the NS units.

## 3 METHOD

The nonlocal operator can be divided into two steps: calculating the affinity matrix $\mathbf{A}$ to represent the correlations between each position pairs and refining the feature map by calculating the

weighted means based on $\mathbf{A}$. In this section, a fully-connected graph filter is utilized for explaining the nonlocal operator. With the Chebyshev approximation, we propose the SNL operator which is proved to be a generalized form of NL and NS operator and is more robust with higher performance in computer vision tasks. Furthermore, based on the stable hypothesis that deeper nonlocal structure tends to learn a stable affinity matrix, we extend our SNL operator into a full-order Chebyshev approximation version, i.e. the gSNL.

## 3.1 THE PROPOSED SPECTRAL NONLOCAL OPERATOR

**Nonlocal operator in the graph view** The nonlocal operator $\mathcal{F}(\mathbf{A}, \mathbf{Z})$ is a filter that computes a weighted mean of all the positions in the feature map $\mathbf{Z}$ based on the affinity matrix $\mathbf{A}$ and then conduct the feature transformation with the kernel $\mathbf{W}$. This is the same as filtering the signal $\mathbf{Z}$ by a graph filter $\mathbf{\Omega}$ in the graph domain defined by the affinity matrix $\mathbf{A}$ (Shuman et al. (2013)). Based on this perspective (Shuman et al. (2013)), we further define the nonlocal operator as:

**Theorem 1.** *Given an affinity matrix $\mathbf{A} \in \mathbb{R}^{N \times N}$ and the signal $\mathbf{Z} \in \mathbb{R}^{N \times C_s}$, the nonlocal operator is the same as filtering the signal $\mathbf{Z}$ in the graph domain of a fully-connected weighted graph $\mathcal{G}$:*

$$\mathcal{F}(\mathbf{A}, \mathbf{Z}) = \mathbf{Z} * g = \mathbf{U}g_\theta(\mathbf{\Lambda})\mathbf{U^T Z} = \mathbf{U\Omega U^T Z}$$
$$\text{with} \quad \mathbf{L} = \mathbf{D}_L - \mathbf{A} = \mathbf{U^T \Lambda U},$$
(4)

*where the graph filter $\mathbf{\Omega} \in \mathbb{R}^{N \times N}$ is a diagonal parameter matrix, i.e. $\mathbf{\Omega} = diag(\omega)$, $\omega = (\omega_1, \omega_2, ..., \omega_n)$. $\mathcal{G} = (\mathbb{V}, \mathbf{A})$ is a fully-connected graph with the vertex set $\mathbb{V}$ and affinity matrix $\mathbf{A}$. $\mathbf{\Lambda} = diag(\{\lambda_1, \lambda_2, ..., \lambda_i, ..., \lambda_N\})$ and $\mathbf{U} = \{\mathbf{u}_1, \mathbf{u}_2, ..., \mathbf{u}_i, ..., \mathbf{u}_N\}$ are the eigenvectors and eigenvalues of the graph laplacian $\mathbf{L}$.*

This definition requires that the graph laplacian $\mathbf{L}$ has non-singular eigenvalue and eigenvector, so the affinity matrix $\mathbf{A}$ should be a symmetric, non-negative, row-normalized matrix. To meet this requirement, the affinity matrix $\mathbf{A}$ can be obtained by the following steps. First, the affinity kernel is used to calculate the matrix $\mathbf{A}$ (we use the dot product with embeded weight matrix $\mathbf{W}_\phi \in \mathbb{R}^{C_1 \times C_s}$ and $\mathbf{W}_\varphi \in \mathbb{R}^{C_1 \times C_s}$ as the affinity kernel, i.e. $\mathbf{A} = (\mathbf{XW}_\phi)(\mathbf{XW}_\varphi)$). Then we make the matrix $\mathbf{A}$ symmetric: $\bar{\mathbf{A}} = \frac{\mathbf{A}^T + \mathbf{A}}{2}$. We normalize the row of $\bar{\mathbf{A}}$ to make it satisfy $d_i = 1$ and having $\check{\mathbf{A}} = \mathbf{D}_A^{-1}\bar{\mathbf{A}}$. For the simplicity, in the following sections the symmetric, non-negative, row-normalized matrix $\check{\mathbf{A}}$ is denoted as $\mathbf{A}$.

**The proposed spectral nonlocal operator** The graph filter $\mathbf{\Omega}$ in Eq. (4) contains $N$ parameters. To simplify it, we use the Chebyshev polynomials which can reduce the $N$ parameters into $k$ ($k \ll N$). For simplicity, we firstly assume that the input $\mathbf{Z}$, the output $\mathcal{F}(\mathbf{A}, \mathbf{Z})$ and the output $\mathcal{F}(\mathbf{A}, \mathbf{Z})$ have only one channel.

Following the similar method as Defferrard et al. (2016), the $k_{st}$-order Chebyshev polynomials is used to approximate the graph filter function $g_\theta(\mathbf{\Lambda})$:

$$\mathcal{F}(\mathbf{A}, \mathbf{Z}) = \sum_{k=0}^{K-1} \theta_k T_k(\mathbf{L}')\mathbf{Z} \quad \text{with} \quad \mathbf{L}' = 2\mathbf{L}/\lambda_{max} - \mathbf{I_n},$$
$$s.t. \quad T_0(\mathbf{L}') = \mathbf{I_n}, \quad T_1(\mathbf{L}') = \mathbf{L}', \quad T_k(\mathbf{L}') = 2\mathbf{L}'T_{k-1}(\mathbf{L}') - T_{k-2}(\mathbf{L}').$$
(5)

Due to $\mathbf{L}$ is a random walk laplacican, the maximum eiginvalue $\lambda_{max}$ satisfies $\lambda_{max} = 2$ which makes $\mathbf{L}' = \mathbf{A}$ (Shuman et al. (2013)). Then Eq. (5) becomes:

$$\mathcal{F}(\mathbf{A}, \mathbf{Z}) = \sum_{k=0}^{K-1} \theta_k T_k(\mathbf{A})\mathbf{Z} = \theta_0 \mathbf{Z} + \theta_1 \mathbf{AZ} + \sum_{k=2}^{K-1} \theta_k T_k(\mathbf{A})\mathbf{Z},$$
(6)

If $k = 1$, the first-order Chebyshev approximation of Eq. (6) becomes:

$$\mathcal{F}(\mathbf{A}, \mathbf{Z}) = \theta_0 \mathbf{Z} + \theta_1 \mathbf{AZ},$$
(7)

where $\theta_0$ and $\theta_1$ are the coefficients for the first and second term which are approximated by learning with SGD. Then, extending Eq. (7) into multi-channel conditions, we can get the formation of our SNL operator:

$$\mathcal{F}_s(\mathbf{A}, \mathbf{Z}) = \mathbf{ZW_1} + \mathbf{AZW_2},$$
(8)

where $F_s(\mathbf{A}, \mathbf{Z})$ is the SNL operator, $\mathbf{W_1} \in \mathbb{R}^{C_s \times C_1}$, $\mathbf{W_2} \in \mathbb{R}^{C_s \times C_1}$. Finally, a residual connection is added with the SNL operator to form the SNL block:

$$\mathbf{Y} = \mathbf{X} + \mathcal{F}_s(\mathbf{A}, \mathbf{Z}) = \mathbf{X} + \mathbf{Z}\mathbf{W_1} + \mathbf{A}\mathbf{Z}\mathbf{W_2}. \tag{9}$$

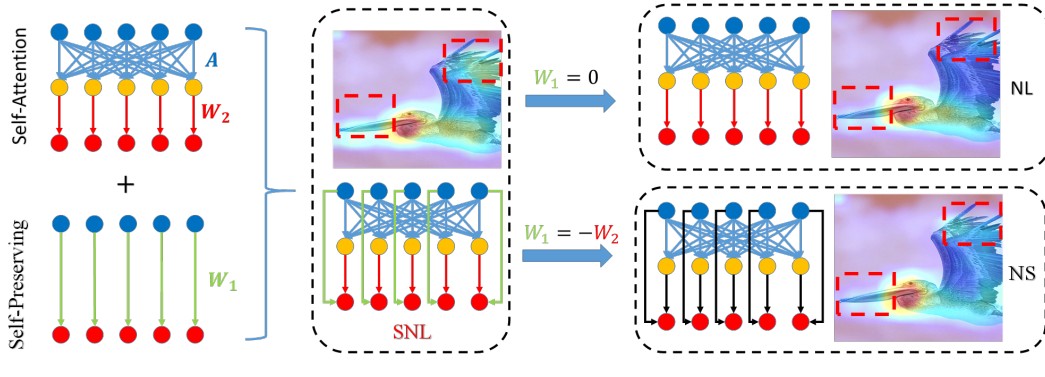

Figure 1: The comparison between the nonlocal operator (NL), nonlocal stage operator (NS) and ours spectral nonlocal operator (SNL). Our SNL has more widely attention range as shown in the two red boxes benefited from the composition of the self-attention term and the self-preserving term which is taken effect by the $\mathbf{W}_1$ and $\mathbf{W}_2$. Our SNL degrades into the NL when $\mathbf{W}_1 = 0$ and NS operator when $\mathbf{W}_1 = -\mathbf{W}_2 = \mathbf{W}$.

**Relation with other nonlocal operators** As shown in fig. 1, our SNL operator can degrade into the NL operator by setting $\mathbf{W_1} = 0$, i.e. $\theta_0 = 0$. However, its analytic solution: $\theta_0 = \frac{2}{N} \sum_{j=0}^{N} \omega_j$ controls the total filtering intensity, which cannot be guaranteed to be 0. This setting will limit the search space when training the network and reduce the robustness of the NL block. The NL operator cannot magnify features of a large range and damp some discriminative features such as the beak of the waterfowl. Our SNL operator can also degrade into the NS operator by setting $\mathbf{W_1} = -\mathbf{W_2}$, i.e. $\theta_1 + \theta_0 = 0$. However, the analytic solution of this equation is $\theta_1 + \theta_0 = \frac{2}{N} \sum_{j=0}^{N} \omega_j(\lambda_j + 1) = 0$. When setting it to zero, the filter strength of the high-frequency signal (with high $\lambda$) such as the small part or twig is suppressed. Thus, it still cannot magnify the discriminative part such as the beak of the waterfowl as shown in fig. 1. Comparing with NL and NS, our SNL does not have these irrational constraints and give these two parameters a liberal learning space. Thus, $\theta_0$ can control the preserve strength of the discriminative features, while $\theta_1$ can pay more attention to the low-frequency signal to diminish the noise.

## 3.2 THE PROPOSED GENERALIZED SPECTRAL NONLOCAL OPERATOR

To fully exploit the "full-range" dependencies, the nonlocal block should have the ability to be consecutively stacked into the network to form a deeper nonlocal structure. However, some types of nonlocal blocks such as the NL and CGNL block cannot achieve this purpose (Tao et al. (2018)). To show the robustness of our SNL block when used in the deeper nonlocal structure, we firstly study the steady-state of deeper nonlocal structure when consecutively adding our SNL block. We also prove the stable hypothesis that the deeper nonlocal structure tends to learn a stable affinity. Based on this hypothesis, we can extend our SNL block into a full-order Chebyshev approximation, i.e. the gSNL block which is more applicable for deeper nonlocal structure.

**The stable hypothesis** The Steady-state analysis can be used to analyze the stable dynamics of the nonlocal block. Here we give the steady-state analysis of our SNL block when consecutively adds into the network structure and get the Stable Hypothesis:

**Lemma 1.** *The Stable Hypothesis: when adding more than two consecutively-connected SNL blocks with the same affinity matrix $\mathbf{A}$ into the network structure, these SNL blocks are stable when the variable affinity matrix $\mathbf{A}$ satisfies: $\mathbf{A}^k = \mathbf{A}$.*

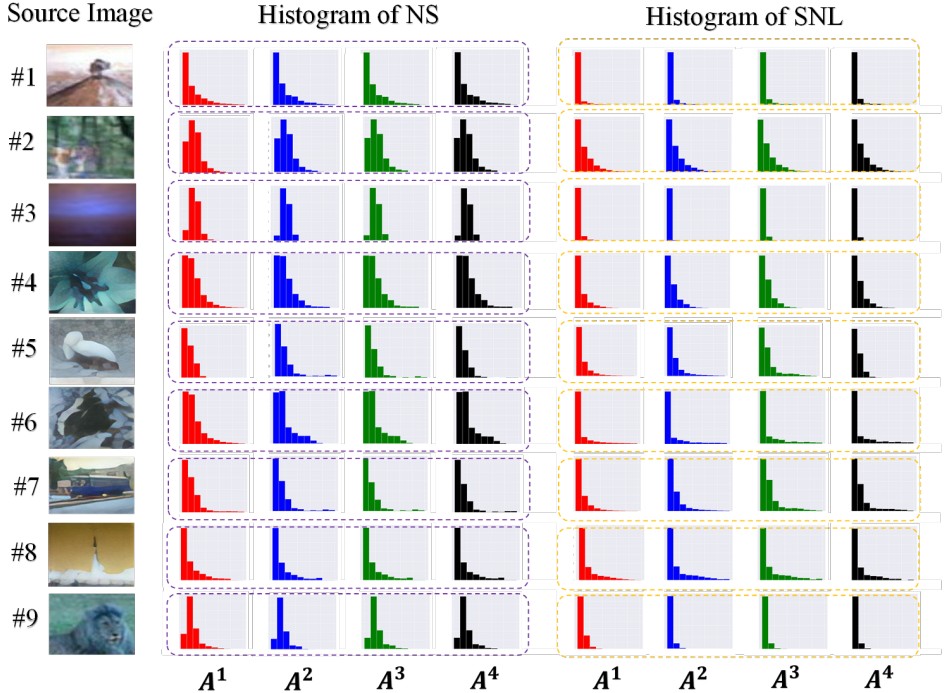

Figure 2: The histogram of the strength statistics of the affinity matrix $A$ where the abscissa is the range of the strength and the ordinates is the number of the elements in $A$ in these ranges. We can see that the histogram of $\mathbf{A}^k$ is nearly the same.

*Proof.* The stability holds when the weight parameters in $W_1, W_2$ and $W$ are small enough such that the *CFL condition* is satisfied (Tao et al. (2018)). So we ignore them for simplicity. The discrete nonlinear operator of our SNL have a similar formulation as the NS operator:

$$\mathcal{L}^h \mathbf{Z}^N := -\mathbf{L}\mathbf{Z},$$

where $h$ is the discretization parameter. $\mathbf{Z}^N$ is the input of the $N^{th}$ block in the deeper nonlocal structure with $\mathbf{Z}^0 = \mathbf{X}$. The stable assumption demands that $\mathbf{Z}^{N+1} = \mathbf{Z}^N$, so the steady-state equation of the last SNL block can be written as:

$$\mathbf{Z}^{N+1} - \mathbf{Z}^N = \mathcal{L}^h \mathbf{Z}^N = -\mathbf{L}\mathbf{Z}^N = 0.$$

The deeper nonlocal structure has more than one SNL blocks. So the $\mathbf{Z}^{N-1}$ and $\mathcal{L}^h \mathbf{Z}^{N-1}$ can be used to express $\mathbf{Z}^N$:

$$-\mathbf{L}\mathbf{Z}^N = -(\mathbf{I} - \mathbf{A})\mathbf{Z}^N = -(\mathbf{I} - \mathbf{A})(\mathbf{Z}^{N-1} + \mathcal{L}^h \mathbf{Z}^{N-1})$$
$$= -(\mathbf{I} - \mathbf{A})\mathbf{Z}^{N-1} + (\mathbf{I} - \mathbf{A})(\mathbf{I} - \mathbf{A})\mathbf{Z}^{N-1} = 0.$$

Finally, the steady-state equation becomes:

$$(\mathbf{I} - \mathbf{A})\mathbf{Z}^{N-1} = (\mathbf{I} - \mathbf{A})^2 \mathbf{Z}^{N-1} \iff \mathbf{A}^2 = \mathbf{A}$$

This equation can naturally extend to the k-hop affinity matrix $\mathbf{A}^k$, i.e. $\mathbf{A}^k = \mathbf{A}$. □

To verify the stable hypothesis, we add five consecutively-connected SNL blocks (and NS blocks) into the PreResnet56 He et al. (2016) and train this model on the train set of the CIFAR100 dataset with the initial learning rate 0.1 which is subsequently divided by 10 at 150 and 250 epochs (total 300 epochs). A weight decay $1e-4$ and momentum 0.9 are also used. Then we test the trained model on the test set and output the affinity matrix of each image. Figure. 2 shows the statistics that reflects the strength of the affinity matrix, 2-hop, 3-hop, and 4-hop affinity matrix: $\mathbf{A}, \mathbf{A}^2, \mathbf{A}^3, \mathbf{A}^4$. We can see that the number of elements in each histogram bin are nearly the same. This means that

the $\mathbf{A}$, $\mathbf{A}^2$, $\mathbf{A}^3$, $\mathbf{A}^4$ have similar distribution of all the elements in k-hop affinity matrixes, which also empirically verifies the stable-state equation: $\mathbf{A}^k = \mathbf{A}$. **Full-order spectral nonlocal operator** With the stable hypothesis, the Chebyshev polynomials can be simplified into a piece-wise function (details in Appendix B). Taking this piece-wise function into the Eq. 7, we can get the full-order approximation of the SNL operator:

$$\mathcal{F}_s^*(\mathbf{A}, \mathbf{Z}) = \sum_k \theta_k T_k(\mathbf{A})\mathbf{Z} = \mathbf{Z}\tilde{\theta}_1 + \mathbf{A}\mathbf{Z}\tilde{\theta}_2 + (2\mathbf{A} - \mathbf{I})\mathbf{Z}\tilde{\theta}_3, \tag{10}$$

where $\tilde{\theta}_1 = \sum_{i_1}^{k\%4=0} \theta_{i_1}$, $\tilde{\theta}_2 = \sum_{i_2}^{k\%4=1||k\%4=2} \theta_{i_1}$, $\tilde{\theta}_3 = \sum_{i_1}^{k\%4=3} \theta_{i_1}$, whose upper bound is less than 1. Then, extending it into multi-channel input and output with the residual connection, we can get our gSNL block:

$$\mathbf{Y} = \mathbf{X} + \mathcal{F}_s^*(\mathbf{A}, \mathbf{Z}) = \mathbf{X} + \mathbf{Z}\mathbf{W}_1 + \mathbf{A}\mathbf{Z}\mathbf{W}_2 + (2\mathbf{A} - \mathbf{I})\mathbf{Z}\mathbf{W}_3 \tag{11}$$

The gSNL block is well-performed when the stable affinity hypothesis is satisfied, i.e. adding more than two nonlocal blocks with the same affinity matrix as shown in Table. 4.

### 3.3 Implementation Details

The implementation details of the gSNL block is shown in fig. 3. The input feature map $\mathbf{X} \in \mathbb{R}^{W \times H \times C_1}$ is first fed into three 1x1 convolutions with the weight kernel: $\mathbf{W}_\phi \in \mathbb{R}^{C_1 \times C_s}$, $\mathbf{W}_\varphi \in \mathbb{R}^{C_1 \times C_s}$, $\mathbf{W}_g \in \mathbb{R}^{C_1 \times C_s}$ to subtract the number of channel. One of the output $\mathbf{Z} \in \mathbb{R}^{W \times H \times C_s}$ is used as the transferred feature map to reduce the calculation complexity, while the other two output $\mathbf{\Phi} \in \mathbb{R}^{W \times H \times C_s}$, $\mathbf{\Psi} \in \mathbb{R}^{W \times H \times C_s}$ are used to get the affinity matrix $\mathbf{A}$. The sub-channel $C_s$ are usually two times less than the input channel $C_1$. The affinity matrix is calculated by the affinity kernel function $f(\cdot)$ and then use the operation in Sec3.1 to make it non-negative, symmetric and normalized. Finally, with the affinity matrix $\mathbf{A}$ and the transferred feature map $\mathbf{Z}$, the output of the nonlocal block can be obtained by the equation Eq. (11). Specifically, the three weight matrixes $\mathbf{W_1} \in \mathbb{R}^{C_s \times C_1}$, $\mathbf{W_2} \in \mathbb{R}^{C_s \times C_1}$, $\mathbf{W_3} \in \mathbb{R}^{C_s \times C_1}$ are implemented as three 1x1 convolutions.

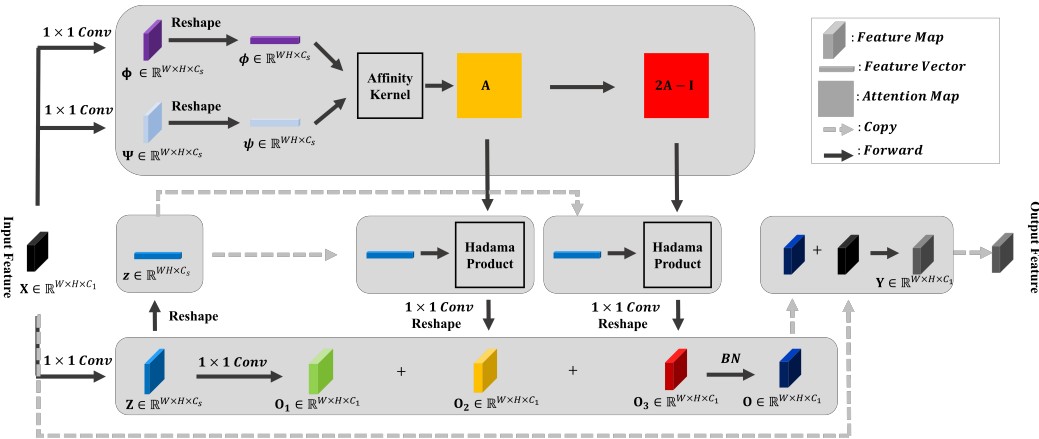

Figure 3: The implementation of the generalized Spectral Nonlocal Block, which added the self-preserving part (green map) and the full-order approximation part (red map) than the NL block.

## 4 Experiment

### 4.1 Setting

**Datasets** Our proposed SNL and gSNL blocks have been evaluated across several computer vision tasks, including image classification and video-based action recognition. For the image classification, both CIFAR-10 and CIFAR-100 datasets (Krizhevsky & Hinton (2009)) are tested. The CIFAR-10 dataset contains $60,000$ images of 10 classes, and CIFAR-100 dataset contains $60,000$ images of 100 classes. For these two datasets, we use $50,000$ images as the train set and $10,000$ images as

the test set. We also generate experiments for the fine-grained classification on the Birds-200-2011 (CUB-200) dataset (Welinder et al. (2010)) which contains $11,788$ images of 200 bird categories. For the action recognition, the experiments are conducted on the UCF-101 dataset (Soomro et al. (2012)), which contains 101 different actions.

**Backbones** For the image classification, the ResNet-50 and the PreResNet variations (including both PreResNet-20 and PreResNet-56) are used as the backbone networks. For the video classification task, we follow the I3D structure (Hara et al. (2018)) which uses $k \times k \times k$ kernels to replace the convolution operator in the residual block.

**Setting for the network** In the main experiments, we set $C_s = C_1/2$. Without loss of the generality, we use the "Dot Product" as the affinity kernel in the experiments. We add one SNL (or gSNL) block into these backbone networks to construct the SNL (or gSNL) network. For the ResNet and the I3D (Hara et al. (2018)), following Wang et al. (2018) we add the SNL block right before the last residual block of $res_4$. For the PreResNet series, we add the SNL block right after the second residual block in $res_1$. For the other nonlocal-base block including the NL (Wang et al. (2018)), the NS (Tao et al. (2018)), the Compact Generalized Nonlocal Block (CGNL) (Yue et al. (2018)) and the Double Attention Block (A2), the settings are all the same as ours. The difference of these blocks are shown in Table. 1, in which the Approximated Condition shows the strategy for the Chebyshev approximation and Channel-wise reflect the consideration of the channel relations.

**Setting for the training** For the image classification on CIFAR-10 dataset and CIFAR-100 dataset, we train the models end-to-end without using pretrained model. The initial learning rate $0.1$ is used for these two datasets with the weight decay $1e-4$ and momentum $0.9$. The learning rate is divided by 10 at 150 and 250 epochs. The models are trained for total 300 epochs.

For the fine-grained classification on CUB-200 dataset, we use the models pretrained on ImageNet (Russakovsky et al. (2015)) to initialize the weights. We train the models for total 200 epochs with the initial learning rate $0.1$ which is subsequently divided by 10 at 31, 61, 81 epochs. The weight decay and momentum are the same as the setting of CIFAR-10 and CIFAR-100.

For the video classification on the UCF-101 dataset, the weights are initialized by the pretrained I3D model on Kinetics dataset (Kay et al. (2017)). We train the models with the initial learning rate $0.1$ which is subsequently divided by 10 each 40 epochs. The training stops at the 100 epochs. The weight decay and momentum are the same as the setting of CIFAR-10 and CIFAR-100.

## 4.2 ABLATION EXPERIMENT

| model | Self-Preserving | Self-Attention | Approximated Conditions | Channel-Wise |
|---|---|---|---|---|
| NL | $\times$ | $\checkmark$ | $\{\theta_i = 0 \mid i = 1, i > 2\}$ | $\times$ |
| A2 | $\times$ | $\checkmark$ | $\{\theta_i = 0 \mid i = 1, i > 2\}$ | $\times$ |
| CGNL | $\times$ | $\checkmark$ | $\{\theta_i = 0 \mid i = 1, i > 2\}$ | $\checkmark$ |
| NS | $\checkmark$ | $\checkmark$ | $\{\theta_i = 0 \mid i > 2\}$ and $\{\theta_1 = -\theta_2\}$ | $\times$ |
| **\*SNL** | $\checkmark$ | $\checkmark$ | $\{\theta_i = 0 \mid i > 2\}$ | $\times$ |
| **\*gSNL** | $\checkmark$ | $\checkmark$ | - | $\times$ |

Table 1: Summary of different types nonlocal block used in the experiments. Our proposed two models have less constraints and are more flexible compared with others.

**The number of channels in transferred feature space** The nonlocal-based block firstly reduces the channels of original feature map $C_1$ into the transferred feature space $C_s$ by the $1 \times 1$ convolution to reduce the computation complexity. When $C_s$ is too large, the feature map will contain redundant information which introduces the noise when calculating the affinity matrix $\mathbf{A}$. However, if $C_s$ is too small, it is hard to reconstruct the output feature map due to inadequate features. To test the robustness for the number of the $C_s$, we generate three types of models with different number of the transferred channels with the setting: "Sub 1" ($C_s = C_1$), "Sub 2" ($C_s = \frac{C_1}{2}$), "Sub 4" ($C_s = \frac{C_1}{4}$) as shown in Table. 2. Other parameters of the models and the training steps are the same as the setting in Sec.4.1. Table. 2 shows the experimental results of the three types of models with different nonlocal blocks. Our SNL and gSNL blocks outperforms other models profited by their flexible for learning. Moreover, from Table. 2, we can see that the performances of the CGNL steeply drops when the number of the transferred channels increases. This is because the CGNL block concerns

the relationship between channels, when the number of the sub-channel increases, the relationship between the redundant channels seriously interferes its effects. Overall, our proposed nonlocal block is the most robust for the large number of transferred channels (our model rise $1.1\%$ in Top1 while the best of others only rise $0.4\%$ compared to the baseline).

Table 2: Experiments for transferred channels on CIFAR100 Dataset

|  | model | top1 | top5 |
|---|---|---|---|
| - | PR-56 | 75.33% | 93.97% |
| Sub 1 | + NL | 75.29% | 94.07% |
|  | + NS | 75.39% | 93.00% |
|  | + A2 | 75.51% | 92.90% |
|  | + CGNL | 74.71% | 93.60% |
|  | + *SNL | **76.34%** | **94.48%** |
|  | + *gSNL | 76.21% | 94.42% |
| Sub 2 | + NL | 75.31% | 92.84% |
|  | + NS | 75.83% | 93.87% |
|  | + A2 | 75.58% | 94.27% |
|  | + CGNL | 75.75% | 93.47% |
|  | + *SNL | **76.41%** | **94.38%** |
|  | + *gSNL | 76.07% | 94.16% |
| Sub 4 | + NL | 75.50% | 93.75% |
|  | + NS | 75.61% | 93.66% |
|  | + A2 | 75.61% | 93.61% |
|  | + CGNL | 75.27% | 93.05% |
|  | + *SNL | 76.02% | 94.08% |
|  | + *gSNL | **76.05%** | **94.21%** |

Table 3: Experiments for different positions on CIFAR100 Dataset

|  | model | top1 | top5 |
|---|---|---|---|
| - | PR-56 | 75.33% | 93.97% |
| Stage 1 | + NL | 75.31% | 92.84% |
|  | + NS | 75.83% | 93.87% |
|  | + A2 | 75.58% | 94.27% |
|  | + CGNL | 75.75% | 93.47% |
|  | + *SNL | **76.41%** | **94.38%** |
|  | + *gSNL | 76.07% | 94.16% |
| Stage 2 | + NL | 75.64% | 93.79% |
|  | + NS | 75.74% | 94.02% |
|  | + A2 | 75.60% | 93.82% |
|  | + CGNL | 74.64% | 92.65% |
|  | + *SNL | **76.29%** | **94.27%** |
|  | + *gSNL | 76.02% | 93.98% |
| Stage 3 | + NL | 75.28% | **93.93%** |
|  | + NS | 75.44% | 93.86% |
|  | + A2 | 75.21% | 93.65% |
|  | + CGNL | 74.90% | 92.46% |
|  | + *SNL | 75.68% | 93.90% |
|  | + *gSNL | **75.74%** | 93.78% |

**The stage for adding the nonlocal blocks** The nonlocal-based blocks can be added into the different stages of the preResNet (or the ResNet) to form the Nonlocal Net. In Tao et al. (2018), the nonlocal-based blocks are added into the early stage of the preResNet to catch the long-range correlations. Here we experiment the performance of adding different types of nonlocal blocks into the three stages (the first, the second and the third stage of the preResNet) and train the models on CIFAR100 dataset with the same setting discussed in Sec.5.2. The experimental results are shown in Table. 3. We can see that the performances of the NL block is lower than the backbones when adding into the early stage. However, our proposed SNL block has $0.81\%$ improvement compared with the backbone when respectively adding into all the three stages, which is much higher than the other type nonlocal blocks (only $0.42\%$ for the best case).

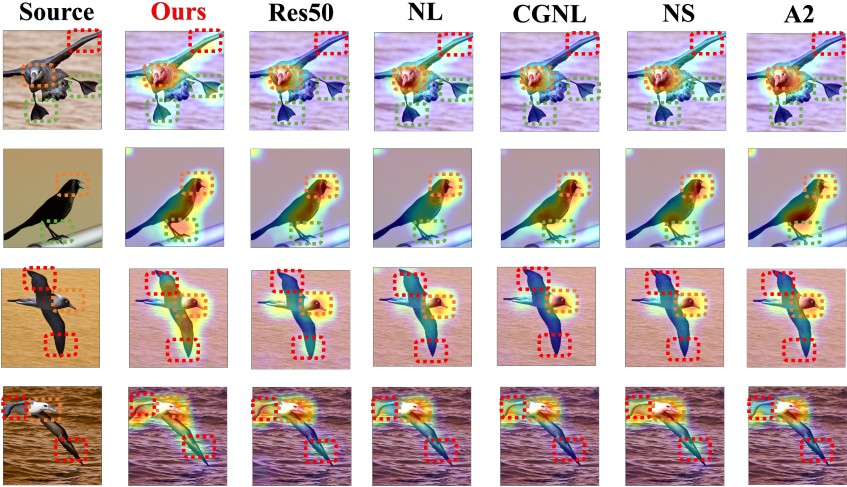

Figure 4: The Feature maps of Nonlocal-based Network. Our SNL block has better results for the crucial part of the birds as shown in the highlighted boxes.

To intuitively show the stability and robustness of our SNL, we give the spectrum analysis for the estimated weight matrices (Tao et al. (2018)). We extract the self-attention weight matrix: $\mathbf{W}_g, \mathbf{W}$ of the NL block and the NS block, $\mathbf{W}_g, \mathbf{W}_2$ of our proposed SNL block. The dimension of the weight matrix satisfies: $\mathbf{W}_g \in \mathbb{R}^{C_1 \times C_s}$, $\mathbf{W} \in \mathbb{R}^{C_s \times C_1}$ $\mathbf{W}_2 \in \mathbb{R}^{C_s \times C_1}$. To make all the eigenvalues real, we let: $\widetilde{\mathbf{W}} = \frac{(\mathbf{W}_g\mathbf{W})+(\mathbf{W}_g\mathbf{W})^T}{2}$. We do the same to the $\mathbf{W}_2$. Figure. 5 shows the top thirty-two eigenvalues of the weight matrix of $\widetilde{\mathbf{W}}$ on the models in Table. 3. We can see that the density of the negative eigenvalues is higher than the positive eigenvalues of the NL block when adding into all three stages. This phenomenon makes the NL operator $\mathcal{F}(\mathbf{A}, \mathbf{Z})$ in Eq. (1) less than zero. So the output feature map is less than the input feature map, i.e. $\mathbf{Y} < \mathbf{X}$ (more detail of this phenomenon can be seen in Tao et al. (2018)). The NS block can avoid "the damping effect" to some extent by concerning the diffusion nature. However, when adding into the early stage, only six eigenvalues of the nonlocal stage are not equal to zero. This phenomenon makes the nonlocal stage cannot effectively magnify the discriminated feature. Comparing with these two models, our proposed SNL block has more positive eigenvalues which takes effect to enhance the discriminated features and also avoids the "damping effect".

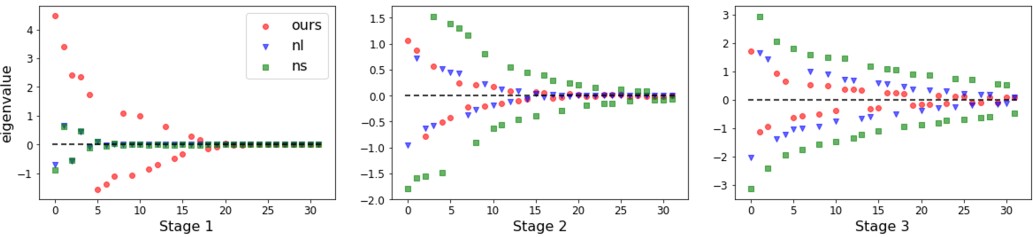

Figure 5: The eigenvalue of the the nonlocal weight matrix trained on the CIFAR100 dataset

**The number of the nonlocal blocks** We test the robustness for adding multiple nonlocal blocks into the backbone network which forms the three type network "Different Position 3 (DP 3)", "Same Position 3 (SP 3)" "Same Position 5 (SP 5)" as shown in Table. 4. The result are shown in Table. 4. For the model "DP3", three blocks are added into the stage 1, stage 2, and stage 3 (right after the second residual block). We can see that adding three proposed nonlocal operators into different stages of the backbone generate a larger improvement than the NS operator and NL operator ($2.4\%$ improvement). This is because when adding NS and NL into the early stage, these two models cannot better aggregate the low-level features and interfere the following blocks. For the model "SP 3" ("SP 5"), we add three (five) consecutively-connected nonlocal blocks into the stage 1. Note that different from the experiment in Tao et al. (2018) and Wang et al. (2018), these consecutively-connected nonlocal blocks have the same affinity matrix. From Table. 4, we can see that profited by concerning the stable hypothesis discussed in Sec 3.3, our gSNL outperform all other models when adding consecutively-connected nonlocal blocks (rises average $0.72\%$ to the backbone and $0.41\%$ higher than the best performance of other type nonlocal blocks) and has a relatively stable performance. However, one drawback is that our gSNL may interfere the learning when adding only one nonlocal block (the stable hypothesis is not satisfied).

### 4.3 MAIN RESULTS

We test the networks with the Nonlocal Block (NL), the Nonlocal Stage (NS), the Compact Generalized Nonlocal block (CGNL), the Double Attention Block (A2) and our SNL (gSNL) blocks in the different visual learning tasks. The experiment settings are discussed in Sec.4.1. Our models outperform other types of the nonlocal blocks across several standard benchmarks. Table. 5 shows the experimental results on the CIFAR10 dataset, we can see that by adding one proposed block, the Top1 rises about $0.65\%$, which is higher than adding other type nonlocal blocks ($0.3\%$). As the experiments on CIFAR100 dataset shown in Table. 7, using our proposed block brings improvement about $1.8\%$ with ResNet50. While using a more simple backbone PreResnet56, our model can still generate $1.1\%$ improvement as shown in Table. 6.

Table. 9 shows the experimental results on the fine-grained image classification task on CUB-200 datasets. Our model outperforms other non-channel-concerning blocks and generate ($0.42\%$) im-

| | model | top1 | top5 | | model | top1 | top5 |
|---|---|---|---|---|---|---|---|
| - | PR-56 | 75.33% | 93.97% | - | PR-56 | 75.33% | 93.97% |
| SP 1/DP 1 | + NL | 75.31% | 92.84% | SP 3 | + NL | 75.43% | 93.67% |
| | + NS | 75.83% | 93.87% | | + NS | 75.30 % | 93.74% |
| | + A2 | 75.58% | 94.27% | | + A2 | 75.23% | 94.03% |
| | + CGNL | 75.75% | 93.47% | | + CGNL | 75.64% | 93.05% |
| | + *SNL | **76.41%** | **94.38%** | | + *SNL | 75.70% | 94.10% |
| | + *gSNL | 76.07% | 94.16% | | + *gSNL | **76.16%** | **94.32%** |
| DP 3 | + NL | 74.34% | 93.31% | SP 5 | + NL | 75.13% | 93.53% |
| | + NS | 75.00% | 93.57% | | + NS | 75.25% | 94.00% |
| | + A2 | 75.63% | 94.12% | | + A2 | 75.61% | 93.81% |
| | + CGNL | 75.96% | 93.10% | | + CGNL | 75.15% | 92.93% |
| | + *SNL | **76.70%** | 93.94% | | + SNL | 76.04% | 94.19% |
| | + *gSNL | 76.45% | **94.53%** | | + gSNL | **76.04%** | **94.35%** |

Table 4: Experiments for different numbers on CIFAR100 Dataset

provement. Comparing with the channel-wise concerning CGNL block, our model is only a bit lower in Top1. Fig. 4 also shows the visualized feature map which is formed by adding the up-sampled feature output with the source image. We can see that the feature maps of our proposed block can cover more critical area of the birds. For example, both the left and right wings (red square) of the birds can be focused profited by the better long-range concerning of our SNL. Moreover, benefited from the flexibility of the $W_1$, our proposed SNL can also catch a relatively large range of the discriminative parts. Table. 8 shows the experimental results on the action recognition task. The network with our proposed block can generate $1.8\%$ improvement than the I3D model and outperforms all other nonlocal models on the UCF-101 dataset.

Table 5: The Results on Cifar10 Table 6: The Results on Cifar100 Table 7: The Results on Cifar100

| model | top1 | top5 |
|---|---|---|
| PR-20 | 94.94% | 99.87% |
| + NL | 94.01% | 99.82% |
| + NS | 95.15% | 99.88% |
| + A2 | 92.44% | 99.86% |
| + CGNL | 94.49% | 99.92% |
| + *SNL | 94.69% | 99.84% |
| + *gSNL | **95.59%** | **99.92%** |

| model | top1 | top5 |
|---|---|---|
| PR-56 | 75.33% | 93.97% |
| + NL | 75.31% | 92.84% |
| + NS | 75.83% | 93.87% |
| + A2 | 75.58% | 94.27% |
| + CGNL | 75.75% | 93.47% |
| + *SNL | **76.41%** | **94.38%** |
| + *gSNL | 76.07% | 94.16% |

| model | top1 | top5 |
|---|---|---|
| R-50 | 76.50% | 93.14% |
| + NL | 76.77% | 93.55% |
| + NS | 77.90% | 94.34% |
| + A2 | 77.30% | 93.40% |
| + CGNL | 74.88% | 92.56% |
| + *SNL | **78.17%** | **94.17%** |
| + *gSNL | 77.28% | 93.63% |

Table 8: The Results on UCF101 Table 9: The Results on CUB

| model | top1 | top5 |
|---|---|---|
| I3D | 81.57% | 95.40% |
| + NL | 81.37% | 95.76% |
| + NS | 82.50% | 95.84% |
| + A2 | 82.68% | 95.85% |
| + CGNL | 83.16% | 96.16 % |
| + *SNL | 82.30% | 95.56% |
| + *gSNL | **83.21%** | **96.53%** |

| model | top1 | top5 |
|---|---|---|
| R-50 | 85.43% | 96.70% |
| + NL | 85.34% | 96.77% |
| + NS | 85.54% | 96.56% |
| + A2 | 86.02% | 96.56% |
| + CGNL | **86.14%** | 96.34% |
| + *SNL | 85.91% | 96.65% |
| + *gSNL | 85.95% | **96.79%** |

# 5 CONCLUSION

In this paper, we explain the nonlocal block in the graph view and propose the spectral nonlocal (SNL) block which is more robust and well-behaved. Our SNL block is a generalized version of the NL and NS block and having more liberty for the parameter learning. We also give the stable hypothesis for deeper nonlocal structure and extend the SNL to gSNL that can be applied to the deeper nonlocal structures. The experiments on multiple computer vision tasks show the high robustness and performance of our proposed nonlocal block. Feature works will focus on using the SNL block into different vision task and its roubustness for the other type of neural network such as the Generative Adversarial Networks (GAN).

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

## A    ANALYTIC SOLUTION OF THE CHEBYSHEV APPROXIMATE

Here we give the analytic solution for the coefficients in Chebyshev polynomials (Phillips (2003)):

**Theorem 2.** *Giving a function $f(\mathbf{x})$, $\mathbf{x} = \{x_1, x_2, ..., x_N\}$, it can be optimally approximated by Chebyshev polynomials: $f(\mathbf{x}) \approx \sum_{k=0}^{K-1} a_k T_k(\mathbf{x})$, only when $a_k$ satisfies: $a_k = \frac{2}{N}\sum_{j=0}^{N} f(\mathbf{x_j})T_k(\mathbf{x_j})$. We call the $a_k$ as the analytic solution of the Chebyshev coeffcients.*

Based on these theorem, we can get the analytic solution of the parameter $\theta$ for Eq. (7):

**Lemma 2.** *The spectral nonlocal operator can be best approximated when the function $g(\lambda) = \omega$ can be best approximated by the Chebyshev polynomials, i.e. the analytic solutions of the Chebyshev coeffcients satisfy:*

$$\theta_k = a_k = \frac{2}{N}\sum_{j=0}^{N} g(\lambda_j)T_k(\lambda_j) = \frac{2}{N}\sum_{j=0}^{N} \omega_j T_k(\lambda_j) \tag{12}$$

## B    THE PIECEWISE CHEBYSHEV POLYNOMIALS

Taking $\mathbf{A}^k = \mathbf{A}$ into the Chebyshev polynomials of the affinity matrix $\mathbf{A}$, the Chebyshev polynomials becomes:

$$\begin{aligned}
T_0(\mathbf{A}) &= \mathbf{I} \\
T_1(\mathbf{A}) &= \mathbf{A} \\
T_2(\mathbf{A}) &= 2\mathbf{A}T_1(\mathbf{A}) - T_0(\mathbf{A}) = 2\mathbf{A}\mathbf{A} - \mathbf{I} = 2\mathbf{A} - \mathbf{I} \\
T_3(\mathbf{A}) &= 2\mathbf{A}T_2(\mathbf{A}) - T_1(\mathbf{A}) = 2\mathbf{A}(2\mathbf{A} - \mathbf{I}) - \mathbf{A} = \mathbf{A} \\
T_4(\mathbf{A}) &= 2\mathbf{A}T_3(\mathbf{A}) - T_2(\mathbf{A}) = 2\mathbf{A}\mathbf{A} - 2\mathbf{A} + \mathbf{I} = \mathbf{I} = T_0(\mathbf{A}) \\
T_5(\mathbf{A}) &= 2\mathbf{A}T_4(\mathbf{A}) - T_3(\mathbf{A}) = 2\mathbf{A}\mathbf{I} - \mathbf{A} = \mathbf{A} = T_1(\mathbf{A}) \\
T_6(\mathbf{A}) &= 2\mathbf{A}T_5(\mathbf{A}) - T_4(\mathbf{A}) = 2*T_2(\mathbf{A}) - T_1(\mathbf{A}) = T_2(\mathbf{A})
\end{aligned} \tag{13}$$

This cyclic form of Chebshev polynomials $T_k(A)$ can be reformulated as a piecewise function:

$$T_k(\mathbf{A}) = \begin{cases} \mathbf{I} & k\%4 = 0 \\ \mathbf{A} & k\%4 = 1 \quad || \quad k\%4 = 3 \\ 2\mathbf{A} - \mathbf{I} & k\%4 = 2 \end{cases} \tag{14}$$

## C    EXPERIMENT OF SEMANTIC SEGMENTATION ON VOC2012 DATASET

For the semantic segmentation tasks, we generate experiment on the VOC2012 dataset with the model proposed by Chen et al. (2017).We add different types of nonlocal blocks on right before the last residual block in $res4$ of the ResNet50. The models are trained for 50 epochs with the SGD optimize algorithm. The learning rate is set $0.007$ with the weight decay $5e-4$ and momentum $0.9$. Experimental results show that the model with our proposed block can the best results.

| model | mIoU | fwIoU | acc |
|-------|------|-------|-----|
| R-50 | 0.713 | 0.868 | 0.926 |
| + NL | 0.722 | 0.872 | 0.927 |
| + NS | 0.722 | 0.873 | 0.927 |
| + A2 | 0.723 | 0.874 | 0.928 |
| + CGNL | 0.722 | 0.872 | 0.928 |
| + *SNL | 0.726 | **0.875** | **0.930** |
| + *gSNL | **0.727** | **0.875** | 0.929 |

Table 10: Experiment on VOC2012 Dataset

## D    THE EXAMPLE OF THE AFFINITY MATRIX ON CUB DATASETS

Experiments to verify the stable hypothesis is also generated on the CUB datasets, we add three consecutively-connected SNL blocks (and NS blocks) into the ResNet50 (right before the last residual block of $res_4$) and train this model on the train set of the CUB dataset with the initial learning rate 0.1 which is subsequently divided by 10 at 31, 61 and 81 epochs (total 200 epochs). A weight decay $1e - 4$ and momentum 0.9 are also used. Figure. 6 shows the histogram of the strength statistics of the affinity matrix $A$. We can see that although using different backbone and dataset, the distribution of the k-hop affinity matrixes are corresponded with the experiments on CIFAR100.

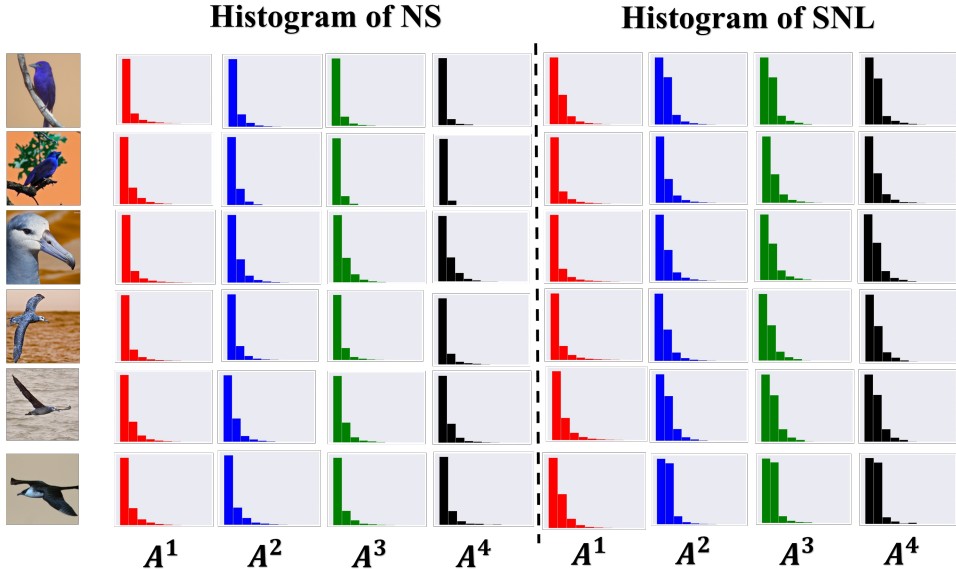

Figure 6: The histogram of the strength statistics of the affinity matrix $A$ where the abscissa is the range of the strength and the ordinates is the number of the elements in $A$ in these ranges. We can see that the histogram of $\mathbf{A}^k$ is nearly the same.

## E    EXPERIMENTS ON VIDEO-BASED PERSON RE-IDENTIFICATION

Experiments are also conducted on the challenging datasets on Video-based Person Re-identification task including the Mars, ILID-SVID and PRID2011. For the backbone, we follow the strategy of Gao & Nevatia (2018) that uses the pooling (RTMtp) and attention (RTMta) to fuse the spatial-temporal features. Note that the models are totally trained on ilidsvid and prid2011 rather than fine-tuning the pre-trained model on Mars dataset. The experimental results are shown in Table.11, 12, 13. We can see that in these datasets, our proposed block can still generate consistent improvements.

Table 11: The Results on Mars dataset

| model | mAP | Rank1 |
|---|---|---|
| RTMta | 77.70% | 79.10% |
| + NL | 72.90% | 80.90% |
| **+ *SNL** | 74.00% | 81.98% |
| RTMtp | 75.70% | 82.30% |
| + NL | 75.54% | 83.40% |
| **+ *SNL** | **76.80%** | **99.92%** |

Table 12: The Results on ILID-SVID dataset

| model | mAP | Rank1 |
|---|---|---|
| RTMta | 69.70% | 58.70% |
| + NL | 66.30% | 56.00% |
| **+ *SNL** | 79.40% | 70.00% |
| RTMtp | 81.60% | 74.70% |
| + NL | 83.00% | 75.30% |
| **+ *SNL** | **84.80%** | **76.60%** |

Table 13: The Results on PRID2011 dataset

| model | mAP | Rank1 |
|---|---|---|
| RTMta | 86.60% | 79.80% |
| + NL | 90.70% | 85.40% |
| **+ *SNL** | 91.50% | 86.50% |
| RTMtp | 90.50% | 86.50% |
| + NL | 89.70% | 85.40% |
| **+ *SNL** | **92.40%** | **88.80%** |

## F  ADDITIONAL EXPERIMENTS ON ACTION CLASSIFICATION

Ours SNL can also improve the performance of other network structures such as the Pseudo 3D Convolutional Network (P3D) (Qiu et al. (2017)), the Motion-augmented RGB Stream (MARS) (Crasto et al. (2019)), the Slow-Fast Network (Slow-Fast) (Feichtenhofer et al. (2019)) and the Video Transformer Network (VTN) (Kozlov et al. (2019)). For P3D and MARS, our SNL block is inserted right before the last residual layer of the res3. For the Slow-Fast, we replace its original NL block with our SNL block. For the VTN, we replace its multi-head self-attention blocks (paralleled-connected NL blocks) with our SNL blocks. The Slow-Fast network are trained end-to-end on the UCF-101 dataset while others use the model pretrained on Kinetic400 dataset and finetuning on the UCF-101 dataset. From Table. 14, We can see that all the performances are improved when adding our proposed SNL model.

Experiments on Kinetics-400 dataset are also given in Table. 15. We can see that inserting SNL block into the Slow-Fast Network can generate 2.1% improvement.

| model | Top1 |
|---|---|
| P3D | 81.23% |
| **P3D + *SNL** | 82.65% |
| SlowFast | 81.17% |
| **SlowFast + *SNL** | 80.54% |
| VTN | 90.06% |
| **VTN + *SNL** | 90.34% |
| MARS | 92.29% |
| **MARS + *SNL** | **92.79%** |

Table 14: Experiment on UCF-101 Dataset

| model | Top1 |
|---|---|
| SlowFast | 77.88% |
| SlowFast + NL | 79.16% |
| **SlowFast + *SNL** | **79.98%** |

Table 15: Experiment on Kinetics-400

