# OpenReview forum: "Spectral Nonlocal Block for Neural Network"
_ICLR.cc/2020/Conference — Reject_

### Official Review · AnonReviewer2 · 2019-10-19
**Official Blind Review #2**

**Rating:** 6

**Review:**

In this paper, authors propose a spectral nonlocal block. First, they re-interpret the nonlocal blocks in a graph view and then use Chebyshev approximation to obtain the spectral nonlocal block which is quite simple by adding a ZW_1 term. Furthermore, they analyze the steady-state to build up a deeper nonlocal structure. Also, the gSNL is simple by adding a (2A-I)ZW_3 term.

Overall, the paper is written well. I like the idea to interpret the nonlocal operation in the graph view. More important, the resulting formulation is quit concise for implementation. However, my main concern is the experiment, which should be further enhanced by perform large-scale video classification like Kinetics400.

**Experience Assessment:**

I have read many papers in this area.

**Review Assessment: Checking Correctness Of Derivations And Theory:**

I assessed the sensibility of the derivations and theory.

**Review Assessment: Checking Correctness Of Experiments:**

I assessed the sensibility of the experiments.

**Review Assessment: Thoroughness In Paper Reading:**

I read the paper at least twice and used my best judgement in assessing the paper.

---

> ### Author Response · Authors · 2019-11-15
> **Add 2 more large-scale video classfications on Mars and Kinetics-400 datasets**
>
> Many thanks for your positive and encouraging comments.
>
> >Improve our experiments on larger datasets.
> We have done experiments on the large-scale video dataset Mars (used for video-person Re-identification), which contains 20,000 videos (nearly 260,000 images) with 1,261 person IDs, the results are shown below:
>
> |                                    |       Mars[1]      |
> |                                    |Rank1 |  mAP   |
> -----------------------------------------------------
> |RTMta                        |79.10%|71.70%|
> -----------------------------------------------------
> |RTMta + NL               |80.90%|72.90%|
> -----------------------------------------------------
> |RTMta + SNL (Ours)|81.90%|74.00%|
> -----------------------------------------------------
> |RTMtp                        |82.30%|75.70%|
> -----------------------------------------------------
> |RTMtp + NL               |83.21%|76.54%|
> -----------------------------------------------------
> |RTMtp + SNL (Ours)|83.40%|76.80%|
> -----------------------------------------------------
>
> *Rank1:  cumulative matching curve at rank-1 (larger is better, maximum 1)
> *mAP: mean average precision score
>
> We can see that in Mars datasets, our proposed block inserted in SOTA networks can generate 1% - 2% improvements consistently. For the network backbone, we follow the strategy of [2] that uses the pooling (RTMtp) and attention (RTMta) to fuse the spatial-temporal features. More details have been included in Appendix E in the updated paper.
>
> Experiments for Kinetics-400 with the state-of-the-art model (slowfast[3]) using our SNL block is shown below, more details have been included in Appendix F in the updated paper.
>
> |    Method                     |  Top1 |
> ----------------------------------------------
> |Slowfast                        |77.88%|
> ----------------------------------------------
> |Slowfast + SNL (Ours)|79.98%|
> ----------------------------------------------
>
> Our SNL inserted into Slowfast model can generate also higher classification accuracy.
>
> In summary, we have done totally 4 computer vision tasks over 9 popular benchmarks, including image classification (Cifar10, Cifar100, CUB-200 in Section4), action recognition (UCF101 in Section4, Kinetics-400 in Appendix F (new added) ), semantic segmentation (VOC2012 in Appendix C), person re-identification (Mars,  ilidsvid, prid2011 in Appendix E (new added)), which all benefit from our SNL. Details please refer to our updated paper.
>
> Thanks again for your comments ! Wish our response make our work more solid for you.
>
>
> [1] Zheng L, Bie Z, Sun Y, et al. Mars: A video benchmark for large-scale person re-identification[C]//European Conference on Computer Vision. Springer, Cham, 2016: 868-884.
> [2] Gao J, Nevatia R. Revisiting temporal modeling for video-based person reid[J]. arXiv preprint arXiv:1805.02104, 2018.
> [3] Feichtenhofer C, Fan H, Malik J, et al. Slowfast networks for video recognition[C]//Proceedings of the IEEE International Conference on Computer Vision. 2019: 6202-6211.

---

### Official Review · AnonReviewer3 · 2019-10-31
**Official Blind Review #3**

**Rating:** 6

**Review:**

SUMMARY:
- Propose spectral non-local block
- improvement on image and video classification tasks

Apologies, I am not at all familiar with the theory and math behind this proposal, I do not think I am in a position to review this paper. The experiments seem convincing enough that the authors made enough effort to prove their method might work.

- Feature maps to show robustness of method is a good point
- CIFAR-10 and CIFAR-100 are certainly a good start, but might not be the best datasets to test for image classification, in lieu of ImageNet and others.
- Classification itself is a good start, it might be interesting to think about using this in a generative model such as GAN. The content reminds me of Self-Attention GAN which uses a similar non-local block (self-attention).

**Experience Assessment:**

I do not know much about this area.

**Review Assessment: Checking Correctness Of Derivations And Theory:**

I did not assess the derivations or theory.

**Review Assessment: Checking Correctness Of Experiments:**

I assessed the sensibility of the experiments.

**Review Assessment: Thoroughness In Paper Reading:**

I read the paper at least twice and used my best judgement in assessing the paper.

---

> ### Author Response · Authors · 2019-11-15
> **18 more experiments on 3 additional datasets have been added to demonstrate our classification results.**
>
> Thanks for reviewing our paper with informative suggestions.
>
> >1. CIFAR-10 and CIFAR-100 datasets may not be the best datasets.
>
> *Additional experiments* on other challenging datasets such as Mars[1], ilidsvid[2], and prd2011[3] have been conducted for video-based person Re-identification tasks as follows. We choose CIFAR-10/100 datasets due to the fairly compared with other SOTAs (Tao et al 2018, He et al, 2019).
>
> |                                    |       Mars[1]      |     ilidsvid[2]     |    prd2011[3]   |
> |                                    |Rank1 |  mAP   | Rank1|  mAP  | Rank1 |mAP   |
> --------------------------------------------------------------------------------------------------
> |RTMta                        |79.10%|71.70%|58.70%|69.00%|79.80%|86.60%|
> --------------------------------------------------------------------------------------------------
> |RTMta + NL               |80.90%|72.90%|56.00%|66.30%|85.40%|90.70%|
> --------------------------------------------------------------------------------------------------
> |RTMta + SNL (Ours)|81.90%|74.00%|70.00%|79.40%|86.50%|91.50%|
> --------------------------------------------------------------------------------------------------
> |RTMtp                        |82.30%|75.70%|74.70%|81.60%|86.50%|90.50%|
> --------------------------------------------------------------------------------------------------
> |RTMtp + NL               |83.21%|76.54%|75.30%|83.00%|85.40%|89.70%|
> --------------------------------------------------------------------------------------------------
> |RTMtp + SNL (Ours)|83.40%|76.80%|76.70%|84.80%|88.80%|92.40%|
> ---------------------------------------------------------------------------------------------------
>
> *Rank1:  cumulative matching curve at rank-1 (larger is better, maximum 1)
> *mAP: mean average precision score
>
> For the backbone, we follow the strategy of [4] that uses the pooling (RTMtp) and attention (RTMta) to fuse the spatial-temporal features. (Note that the models are totally trained on ilidsvid and prid2011 rather than fine-tuning the pre-trained model on *Mars* datasets. We can see that in these datasets, our proposed block can generate consistent improvements on these datasets.
>
> We have added these additional experiments into Appendix E.
>
> [1] Zheng L, Bie Z, Sun Y, et al. Mars: A video benchmark for large-scale person re-identification[C]//European Conference on Computer Vision. Springer, Cham, 2016: 868-884.
> [2] Wang T, Gong S, Zhu X, et al. Person re-identification by video ranking[C]//European Conference on Computer Vision. Springer, Cham, 2014: 688-703.
> [3] Hirzer M, Beleznai C, Roth P M, et al. Person re-identification by descriptive and discriminative classification[C]//Scandinavian conference on Image analysis. Springer, Berlin, Heidelberg, 2011: 91-102.
> [4] Gao J, Nevatia R. Revisiting temporal modeling for video-based person reid[J]. arXiv preprint arXiv:1805.02104, 2018.
>
> >2. Think about using this in a generative model such as GAN.
>
> Thanks for the suggestion! Our nonlocal block has the potential to be applied to the self-attention GAN. More than considering high-resolution details as a function of only spatially local points in lower-resolution feature maps, our SNL is good at generating more details using cues from all feature locations even much better than the conventional nonlocal block (Wang et al 2018). It can thus learn specific structures and geometric features besides only texture features. We have added the future application of the proposed block in the Conclusion part.
>
> Thanks again for your review, and hope our response make the work clearer.

---

### Official Review · AnonReviewer4 · 2019-11-03
**Official Blind Review #4**

**Rating:** 3

**Review:**

The paper proposes a spectral non-local block, which is a generalized method of the non-local block and non-local stage in the literature. The proposed spectral non-local block can be plugged into a neural network to improve its effectiveness. The paper also provides theoretical analyses of the stability of the proposed method, and also extend the method by including more  Chebyshev polynomial terms. Experiments are conducted on image classification and action recognition tasks, and they valid the effectiveness of the proposed method.

The idea is well-motivated, and it is a generalization of existing works in the literature. I do like this idea. However, I am afraid that the idea is not well explained and supported, thus I gave a weak reject to encourage the authors to further improve the paper.

The major concern I have is the reasonability of the experiments.  The experiments in the paper show relative performance gain with respect to a baseline method. It seems that there is a lack of comparison with state-of-the-art methods in the literature. For example, in Table 8, a performance gain is observed when compared with I3D. However, the recent STOA models can achieve much higher accuracy than the baseline. and also the proposed method. Since the proposed method is generic to all neural nets, it makes more sense to compare with SOTA and make improvements based on SOTA.  What is the conclusion from Table 4? Are you trying to demonstrate that the best configuration is DP3, and increasing the number of consecutive non-local blocks (from SP3 to SP5) doesn't work? It is awkward since the paper gives a stable hypothesis for deeper nonlocal structure, but experimentally the deeper structure doesn't work well. Figure 4 is abrupt without much background descriptions. Are the images randomly chosen? Ours here means SNL or gSNL? Is the colored superimposition the attention map (I believe so but the paper doesn't indicate so) and how to interpret it? What is the relation of the coverage of the critical parts on birds and the long-range dependency? More background descriptions and interpretations of the results are needed.

Another concern I have is the clarity of the writing. There are quite a number of informal use of English, mismatched descriptions, undefined acronyms, etc. For example, in the caption of Fig. 1, it is said self-attention and self-preserving are taken effect by W1 and W2, which is contradictory to what is illustrated in the figure. Also, the terms self-attention and self-preserving, and other terms such as CGNL, A2, Hadama (Hadamard?) product, are not formally defined or described.  A lot of grammar errors and informal use of English are present, such as "which lead to", "the weight means", "when using in the neural network", "fig. 4", "Figure. 2", "more liberty for the parameter learning.", etc.

**Experience Assessment:**

I have published in this field for several years.

**Review Assessment: Checking Correctness Of Derivations And Theory:**

I assessed the sensibility of the derivations and theory.

**Review Assessment: Checking Correctness Of Experiments:**

I assessed the sensibility of the experiments.

**Review Assessment: Thoroughness In Paper Reading:**

I read the paper at least twice and used my best judgement in assessing the paper.

---

> ### Author Response · Authors · 2019-11-15
> **Compared with more SOTA methods and improve the clarity of the writing (1/2)**
>
> Thanks so much for pointing out our cons in explaining our idea. We agree with your concerns about reasonability of the experiments.  Hope our explanations below and the updated manuscript can make it clearer to you.
>
> >1. Experiments for adding our block into more SOTA models.
>
> We have done *additional experiments* on more SOTA models inserted with our SNL blocks, the results are shown below:
>
> -----------------------------------------------------------
> |          Methods                   |      Top1      |
> |  P3D [1]                          |     81.23%   |
> |  P3D + SNL(Ours)          |    82.65%    |
> -----------------------------------------------------------
> |  SlowFast [2]                  |     80.54%    |
> |  SlowFast + SNL (Ours) |    83.92%    |
> ----------------------------------------------------------
> |  MARS (RGB) [3]             |    92.29%    |
> |  MARS + SNL (Ours)      |    92.79%    |
> ----------------------------------------------------------
> |  VTN [4]                          |    90.06%    |
> |  VTN + SNL (Ours)         |    90.34%    |
> -----------------------------------------------------------
>
> For Pseudo 3D Convolutional  Network (P3D) and Motion-augmented RGB Stream (MARS) , our SNL block is inserted into the Pseudo- 3D right before the last residual layer of the res3. For the Slow-Fast Network (Slow-Fast), we replace original NL block with our SNL block. For the Video Transformer Network (VTN), we replace its multi-head self-attention blocks (paralleled-connected NL blocks) with our SNL blocks. The slowfast network are trained end-to-end on the UCF-101 dataset while others use the model pretrained on Kinetic400 dataset and finetuning on the UCF-101 dataset.
>
> We can see that all the performances are improved when adding our proposed SNL model. In sum, our SNL blocks have shown superior results across three additional SOTAs (the P3D, SlowFast, VTN and MARS) in the action recognition tasks. We have included these results in Appendix F.
>
> [1] Qiu Z, Yao T, Mei T. Learning spatio-temporal representation with pseudo-3d residual networks[C]//ICCV 2017: 5533-5541.
> [2] Feichtenhofer C, Fan H, Malik J, et al. Slowfast networks for video recognition[C]//ICCV 2019: 6202-6211.
> [3] Crasto N, Weinzaepfel P, Alahari K, et al. MARS: Motion-Augmented RGB Stream for Action Recognition[C]//CVPR 2019: 7882-7891.
> [4] Kozlov A, Andronov V, Gritsenko Y. Lightweight Network Architecture for Real-Time Action Recognition[J]. arXiv preprint arXiv:1905.08711, 2019.
>
> >2. the conclusion from Table 4; Are you trying to demonstrate that the best configuration is DP3, and increasing the number of consecutive non-local blocks (from SP3 to SP5) doesn't work? the paper gives a stable hypothesis for deeper nonlocal structure, but experimentally the deeper structure doesn't work well.
>
> The conclusion of the Table 4 are:
>
> 1. Adding more SNL block into different layers is better than adding into only one layer.  (According to "1" (DP1) and DP3);
> 2. The proposed gSNL block is more robust without performance drops when going deeper based on our stable hypothesis. (According to "1"(SP1), SP3 and SP5).
>
> For clarifying these two conclusions, we replace the "1" into "SP1 or DP1" and  add the explanations in Sec.4 of the updated version.
>
> gSNL is stable and has the potential for constructing the deeper connections. The reason why more blocks do not improve much is actually task dependent. Deeper structures should be able to learn better representations and converge to the optimal solution, while not decreasing the performance (that is why Resnet is proposed for enabling much deeper network structures for learning representations). On one hand, we show the deeper nonlocal structure under stable hypothesis does not have performance drop (SP5 vs SP3), and on the other hand, SP3 (the deeper structure) is better than SP1 (one block), which already validates deeper structure is better. These claims have been updated in Sec 4.
>
> >3. Fig 4 needs background descriptions. Are the images randomly chosen? Ours here means SNL or gSNL? Is the colored superimposition the attention map and how to interpret it?
>
> Fig.4 shows the feature map (not the attention map) of the SNL block, which is the output (e.g. 32 * 32) of the last feature layer up-sampled into the original size (e.g. 512 * 512) of the input image and then added on the source images. Thus, the feature map = up-sampled feature (32*32 -> 512*512)+ source image (512 *512). They are randomly chosen. Both SNL and gSNL are our proposed nonlocal blocks. SNL focuses on more crucial part of the birds benefited from the flexibility of $W_1$ ( $W_1$ controls the intensity of the graph filter, which is related to the feature representations as discussed in Filx et al 2019). We have added more descriptions into Section4.3.

---

> > ### Author Response · Authors · 2019-11-15
> > **Compared with more SOTA methods and improve the clarity of the writing (2/2)**
> >
> >
> > >4. The relation of the coverage of the critical parts on birds and the long-range dependency. More background descriptions and interpretations of the results are needed.
> >
> > The influence of the long-range dependencies on the convergence of critical parts can be shown especially by the similar parts (e.g. the left and right bird feet, left and right bird swing). If the long-range dependency is not well considered, these similar parts of the birds sometimes are not learned simultaneously. For example, in the third row of Fig4, the ResNet only focuses on the right swing of the bird while neglecting the same important left swing. But when adding SNL to concern the long-range dependencies, it can also focus on the left swing, which is also the critical part the same as the right swing. The long-range dependence of similar features are kept via our SNL.
> >
> > >5. Clarity of the writing.
> > We do thank your suggestions about our writing. The informal use of English, mismatched descriptions, undefined acronyms have been solved in the updated paper. All the corrections should improve the writing quality of this paper. In our modified version, we have used the full name of the model for clarity such as the Compact Generalized Nonlocal Block (CGNL), the Double Attention Network (A2Net).
> >
> > Hope our *additional experiments* on 4 extra SOTA models eliminate your concern and we humely ask whether you can improve your rating of our work. Thank you !

---

### Official Review · AnonReviewer1 · 2019-11-04
**Official Blind Review #1**

**Rating:** 1

**Review:**

I have two general concerns, the first is related to the presentation and the second to the relevance of the results.

(1) Presentation is confusing at many points, for instance:

* It is unclear if theorem in Eq. 4 is original or belongs to Shuman et al 2013. (no proof is given)

* Eq. 8 seems an arbitrary decomposition of the original NonLocal operator that could have been proposed without any reference to the Chebyshev expansion (which, on the other hand, is truncated to 1st order with no extra explanation).

* The point of Fig. 1 and Fig 4 is not clear. Fig. 1 explains how SpectralNonLocal reduces to NonLocal and NonLocalStage, but we can see this from the formulas. I dont see how this discussion on the Ws relates to the regions highlighted in the bird.
The same applies to Fig. 4. What are we supposed to see in Fig. 4 (and, more importantly, why?).

* What is the CFL condition? (is it the Courant-Friedrichs–Lewy sampling condition?). How is that related to the values of Ws. Can we take those arbitrarily small as suggested in that proof?

* The upper limit in the sums after Eq. 10 is unclear.

* First time table 4.2 is cited there is no context to understand it. (actually there is no table labeled as "Table 4.2") Where do we see the different number of NonLocal units?. This is only clear when you arrive and read text of page 9 (but not when cited the first time from page 6).

* Explanation of experiments is a little bit confusing (e.g. what does it mean top1 and top5 in tables?). The only explanation of "top-something" I found in the text has to do with eigenvectors in fig. 5. This also apply to the "topX" in the figures?

(2) Nevertheless, the main concern is the scarce relevance of the results: differences of behavior in all tables are about 1%. Then, what is the real advantage of the proposed modification?


**Experience Assessment:**

I have read many papers in this area.

**Review Assessment: Checking Correctness Of Derivations And Theory:**

I assessed the sensibility of the derivations and theory.

**Review Assessment: Checking Correctness Of Experiments:**

I assessed the sensibility of the experiments.

**Review Assessment: Thoroughness In Paper Reading:**

I read the paper at least twice and used my best judgement in assessing the paper.

---

> ### Author Response · Authors · 2019-11-15
> **Presentation of the paper has been improved and Results improvement is not incremental at all (1/2)**
>
> Thank you for your comments. We address specific concerns as follows.
> (1) Related to the presentation
>
> >1. The theorem in Eq.4 is original or belongs to Shuman et al 2013.
>
> The Eq.4 is originally proposed by us, which is derived from the definition of the graph filter in Shunman et a. 2013. In Sec.3.A of this reference, the author gives the formulation of graph filter and demonstrate that it can be used to implement the nonlocal means filter without giving a specific formulation as ours.
>
> Our Eq.4 formulates a mathematical description: $F(A,Z)=U \Omega U^{T} Z$ that defines a fully connected graph filter to represent the nonlocal means. $Z$ is the transformed feature map, $A$ is the affinity matrix, $U$ is the eigenvector and $\Omega$ is the parameter matrix to reflect the strength of the filter.
>
> We have clarified more details about the originality of Eq.4 and its extension from Shuman et al 2013 paper in the first paragraph of Sec.3.1.
>
> >2. The Eq.8 seems to be an arbitrary decomposition of the original nonlocal operator without any reference to the chebyshev expansion.
>
> It is not the arbitrary decomposition of the original nonlocal operator.
>
> Original nonlocal operator (NL) is $F(A,Z)= A Z W$, in which the weight matrix $W$ has dimension: $C_1 \times C_2$ and $A$ has dimension: $N \times N$ ($W$ cannot multiplicate the affinity matrix $A$ directly); our proposed SNL operator is $F_s(A,Z)=Z W_1 + A Z W_2$, thus the first term $Z W_1$ cannot be obtained except for A = I (identity matrix). Our SNL is obtained by approximating the fully-connected graph filter with the help of $1_{st}$-order Chebyshev approximation (Defferrard et al 2016). We have further explained this point in Sec.3.1 of our paper.
>
> >3. Fig1 & 4 are unclear and how the weights variants are related to the regions highlighted in the bird images.
>
> In Fig 1 & 4, the representation of bird wings and beak have been better learned using SNL (with more distinguished regions highlighted), because $W_1$ in our SNL is more flexible without assumptions as in NL and NL-stage.
>
> Fig.4 gives more examples of the feature maps, which shows that our proposed SNL focuses on more crucial part of the birds (the same as fig 1) benefited from the flexibility of $W_1$ ($W_1$ controls the intensity of the graph filter, which is related to the feature representations as discussed in Filx et al 2019)
>
> We have added the retrospects of why the proposed novel operators have achieved better discriminative feature representations in the second paragraph of Sec.4.3.
>
> >4. The explanation of CFL; how is it related to the values of Ws; Can we take very small ?
>
> Yes, CFL condition is the Courant-Friedrichs-Lewy sampling condition.  This condition holds when the weight parameters are small, which has been demonstrated in Tao et al 2018. A brief illustration here, if using the same affinity matrix $A$ and connecting multiple SNL blocks, these successive blocks can be seen as a diffusion progress which satisfies: $$X^{N + 1} - X^{N}=\frac{dX}{dt} = X N W_{1}+ A X^{N} W_{2} \quad, s.t.\quad a_{ij} > 0, x_{ij} > 0$$
>
> If $|W_{1}|$ and $|W_{2}|$ are much larger than 0, i.e. $|W_{1}| \gg 0 $, $|W_{2}| \gg 0$. It will make $|X^{N} W_{1}| \gg 0$, $|A X^{N} W_{2}| \gg 0$ and then make $|\frac{dX}{dt}| \gg 0$ which does not satisfies the CFL condition and lead to the unstable dynamics, details can be seen in Tao et al 2018. We also verified the authenticity of this hypothesis in the following table by considering the learned weight parameters in $W_{1}$ and $W_{2}$:
>
> |The range of weight valus|  <-0.4  | (-0.4,-0.2)|  (-0.2, -0.1)  |  (-0.1, 0)  |  (0, 0.1)  |  (0.1, 0.2) |  (0.2,0.4)  |  >0.4   |
> |Number of weights           |    14     |    151      |      10104     |  526557 |  525300 |    10099   |     156       |    17    |
> |Percentage                         |0.001%|  0.014%  |      0.942%   |    49.1%  |  49.0%   |   0.942%   |   0.014%  |0.001%|
>
> We can see that nearly 98% of learned weights are in the range of (-0.1, 0.1). The maximum and minimum value of those weight parameters are -0.72 and 0.82, which meet the CFL condition.
>
> The above results also reflect that we cannot take arbitrarily small value for the weight parameters matrix $W$, because there are still some parameters in the range of (-0.2, -0.1) and (0.1, 0.2).
>
> >5. The upper limit in the sums after Eq. 10 is unclear.
>
> The sums after Eq.10 are the learned weight parameters which form the matrix $W_{1}$, $W_{2}$, so it should also satisfy the CFL condition as we discussed above.  From the above table, the empirical study shows that the sums are less than 1. Sorry that we can only prove the sums should meet CFL condition while cannot in theory prove its upper bound (less than "1") at current stage.
>
> We have already clarified these points after the Eq.10 in our updated version.

---

> > ### Author Response · Authors · 2019-11-15
> > **Presentation of the paper has been improved and Results improvement is not incremental at all (2/2)**
> >
> > >6. Explanation of table 4.2. Where do we see different number of nonlocal units.
> >
> > Sorry for the typo. Table.4.2 is actually the Table.4 mentioned in the context, we have corrected it in the updated version.
> >
> > The experimental results of different numbers are shown in Table.4. According to the results of DP1 and DP3, we can see that inserting our SNL block into more layers can achieve higher performance than other models and has better results than inserting it into only one layer. Moreover, the proposed gSNL block are more stable learned than others by meeting our proposed stable hypothesis based on the results of SP1, SP3, SP5.
> >
> > >7. Explanations of Top1 and Top5 accuracy in the paper.
> >
> > Top1 and Top5 is the evaluation criterion of the image classification and action classification. Top1 accuracy means that the model prediction (the one with the highest probability) is exactly the expected label. Top5 accuracy means that any of your model that gives 5 highest probability answers that match the expected answer. Sorry for the confusion, we have added details for distinguishing from  the confused word “top 32 eigenvalues” at the first paragraph of page 9.
> >
> > (2) Related to the results.
> >
> > The spectral nonlocal (SNL) block is an efficient, simple, and generic component for capturing long-range spatial-temporal dependencies with deep neural networks. Compared with other alternatives (NL, CGNL NL-stage), we focus on emphasizing both effciency and robustness of SNL w.r.t. the number and position inserted in the network, which have been shown in Table.2,3, and 4.
> >
> > Because the large-scale datasets are used for validating our block, the improvement 1% is much higher than alternatives. The CGNL (Yue et al 2018) only improves 0.58% and the NS (Tao et al 2018) improves 0.09% than the original NL block.
> >
> > To eliminate your concern, we have also done *additional experiments* on the video-based person re-identification task, which utilizes the network structure RTM with attention called RTMta (Gao et al 2018). It shows that our proposed block can generate a clear-cut improvement (nearly 14% accuracy improvement on the ilidsvid dataset)
> >
> > |     Methods                |     ilidsvid          |
> > |                                    |Rank1 |  mAP   |
> > -------------------------------------------------------------
> > |RTMta                         |58.70%|69.00%|
> > -------------------------------------------------------------
> > |RTMta + NL                |56.00%|66.30%|
> > -------------------------------------------------------------
> > |RTMta + SNL (Ours)   |70.00%|79.40%|
> > -------------------------------------------------------------
> >
> > *Rank1:  cumulative matching curve at rank-1 (larger is better, maximum 1)
> > *mAP: mean average precision score
> >
> > We have included this person re-identification task results in Appendix E. Furthermore, we would like to note that the performance improvements are actually task-dependent. We have also summarized that the different tasks we have done: image classification (datasets: Cifar10, Cifar100, CUB-200 in Section4), action recognition (datasets: UCF101 in Section4, Kinetics-400 in Appendix F (new added) ), semantic segmentation (datasets: VOC2012 in Appendix C), person re-identification (datasets: Mars,  ilidsvid, prid2011 in Appendix E (new added)) benefit differently from our SNL, but they consistently have improvements.
> >
> > We here provide the details of the improvements of all the nonlocal based methods, which are mostly less or around 1%.
> >
> > -----------------------------------------------------------------------------------------------------------------------
> > |                                                       | CIFAR10|CIFAR100| CUB200| UCF101|   Mars |   ilidsvid  |prid2011|VOC2012| Kinetics-400  |
> > | NS (Tao et al NeurIPS 2018)       | + 0.09% |  +0.09%  |       -      |      -        |       -    |       -       |        -       |        -        |        -     |
> > | CGNL (Yue et al, NeurIPS 2018) |       -       |       -         | +0.58% | +1.03% |       -     |        -      |        -      |       -          |  +1.20%     |
> > | A2 (Chen et al, NeurIPS 2018)   |       -        |      -         |      -        |  +0.04%|      -       |       -       |       -       |        -        |  +2.6%     |
> > | CGD (He et al,arXiv:1907.09665)|       -      | +0.99%   |      -       |  +1.18% |      -        |       -        |       -       |        -       |   +0.3%     |
> > |               Ours                               |  +1.58% |  +1.10%  | +0.56% |  +1.84% | +1.00% | +7.35%  |  +2.75%|  +0.3%  |   +0.82%   |
> >
> > * '-' : means that the dataset are not tested in their paper.
> > * +' : means the average improvement compared with the original NL block in each dataset.
> >
> > Many thanks for your time and suggestions, and hope our responses and updated paper make our presentation and results more solid to you.

---

### Decision · Program_Chairs · 2019-12-19

**Decision:**

Reject

**Comment:**

This paper proposes a new formulation of the non-local block and interpret it from the graph view. The idea is interesting and the experimental results seems to be promising.

Reviewer has two major concerns. The first is the presentation, which is not clear enough. The second is the experimental design and analysis. The authors add more video dataset in the revision, but still lack comprehensive experimental analysis for video-based applications.

Overall, the idea of non-local block from graph view is interesting. However, the presentation of the paper needs further polish and thus does not meet the standard of ICLR